# Resting-State EEG Functional Connectivity in Children with Rolandic Spikes with or without Clinical Seizures

**DOI:** 10.3390/biomedicines10071553

**Published:** 2022-06-29

**Authors:** Min-Lan Tsai, Chuang-Chin Wang, Feng-Chin Lee, Syu-Jyun Peng, Hsi Chang, Sung-Hui Tseng

**Affiliations:** 1Division of Pediatric Neurology, Department of Pediatrics, Taipei Medical University Hospital, Taipei Medical University, Taipei 110301, Taiwan; 151009@h.tmu.edu.tw (M.-L.T.); lfc2222@yahoo.com (F.-C.L.); jamesc1208@gmail.com (H.C.); 2Department of Pediatrics, School of Medicine, College of Medicine, Taipei Medical University, Taipei 110301, Taiwan; 3School of Medicine, College of Medicine, Taipei Medical University, Taipei 110301, Taiwan; b101108093@tmu.edu.tw; 4Professional Master Program in Artificial Intelligence in Medicine, College of Medicine, Taipei Medical University, Taipei 110301, Taiwan; 5Department of Physical Medicine & Rehabilitation, Taipei Medical University Hospital, Taipei 110301, Taiwan; m003089010@tmu.edu.tw; 6Department of Physical Medicine & Rehabilitation, School of Medicine, College of Medicine, Taipei Medical University, Taipei 110301, Taiwan

**Keywords:** benign childhood epilepsy, centrotemporal spikes, rolandic spikes, graph theory, EEG functional connectivity, focal epilepsy, children

## Abstract

Alterations in dynamic brain network function are increasingly recognized in epilepsy. Benign childhood epilepsy with centrotemporal spikes (BECTS), or benign rolandic seizures, is the most common idiopathic focal epilepsy in children. In this study, we analyzed EEG functional connectivity (FC) among children with rolandic spikes with or without clinical seizures as compared to controls, to investigate the relationship between FC and clinical parameters in children with rolandic spikes. The FC analysis based on graph theory and network-based statistics in different frequency bands evaluated global efficiency, clustering coefficient, betweenness centrality, and nodal strength in four frequency bands. Similar to BECTS patients with seizures, children with rolandic spikes without seizures had significantly increased global efficiency, mean clustering coefficient, mean nodal strength, and connectivity strength, specifically in the theta frequency band at almost all proportional thresholds, compared with age-matched controls. Decreased mean betweenness centrality was only present in BECTS patients with seizures. Age at seizure onset was significantly positively associated with the strength of EEG-FC. The decreased function of betweenness centrality was only presented in BECTS patients with clinical seizures, suggesting weaker local connectivity may lower the seizure threshold. These findings may affect treatment policy in children with rolandic spikes.

## 1. Introduction

Benign childhood epilepsy with centrotemporal spikes (BECTS), or benign rolandic seizures, is a type of common clinical idiopathic focal epilepsy in children and is characterized by easily medically controlled and spontaneous resolution before age 16 [1]. Normal findings on magnetic resonance imaging (MRI) is a diagnostic criterion of BECTS, and most pediatric patients with BECTS visit clinics for sleep-related seizures. Moreover, rolandic spikes are often found incidentally on electroencephalography (EEG) in children without clinical seizures and may impair intellectual capacity [2]. BECTS is considered benign because of the absence of neurological deficits and self-limited; however, some children can present with neuropsychological impairment as well as cognitive and academic problems [2,3,4].

Focal epilepsy is a disorder of the brain network that is characterized by abnormal neuronal excitation and connectivity with synchronization and propagation [5]. Rolandic spikes in BECTS often indicate a bipolar field, including a negative potential field that is maximal at the mid-temporocentral region with a simultaneous dipole in the frontal region that is more abundant during sleep states [6,7]. Recent progress in the acquisition of EEG data has advanced the understanding of alterations in dynamic brain network function in epilepsy. Thus, specific epileptic syndromes may present through a specific pathway or network and further affect cognitive outcomes. The mechanisms of dynamic connectivity in children with BECTS and their association with seizures and cognitive function remain controversial. 

Graph theory is a mathematical tool that is often used in the analysis of functional connectivity of functional MRI (fMRI), electroencephalography (EEG), and magnetoencephalography (MEG). EEG signals are directly generated by neuronal activity. In graph theory, the specific brain regions or electrodes are the nodes of the network, which are connected by edges and can be used to identify the significance or centrality of individual nodes in the brain network [8]. Graph analysis quantifies topographic parameters and properties, specifically the integration and segregation of the EEG networks that influence the overall or local efficiency in the graph [9], and thereby facilitates reconfiguration of both structural and functional connections in various brain disorders [10]. Seizure disorders, especially focal-onset epilepsy, may present not only abnormal recurrent excitation circuits, but also abnormal synchronization due to densely connected network hubs within the brain. Furthermore, changes in dynamic connectivity and pathway characteristics may play a potential role in the pathogenesis of the disease [8,11].

It is unclear whether the EEG connectivity differs in BECTS patients with or without clinical seizures, and the treatment of BECTS patients remains controversial due to the emerging issue of cognitive involvement [3]. In our research, we use four parameters to evaluate functional connectivity at different proportional thresholds. Global efficiency and nodal strength are designated to evaluate the integration of functional networks. Clustering coefficients often reflect the segregation function. Betweenness centrality is a measure of prioritizing nodes situated on the shortest paths in the network to identify the centrality of individual nodes in the network [8]. We hypothesize that BECTS patients with seizures may have different resting-state functional connectivity on EEG compared to those without evident seizures. Our research objectives were to investigate: (1) EEG functional connectivity in the resting state among pediatric patients with BECTS with seizures before pharmacotherapy, children with rolandic spikes without seizures, and age-matched controls, and (2) the relationship between clinical parameters, including cognitive function and EEG network properties.

## 2. Materials and Methods

### 2.1. Clinical Demographics 

We retrospectively collected the medical records of children (age < 18 years) with a history of seizure disorders whose clinical data and results of regular EEGs at Taipei Medical University Hospital (TMUH) between December 2012 and November 2021 were available. BECTS or rolandic spikes were diagnosed by pediatric neurologists and re-evaluated for this research (M.-L.T. and H.-C.) according to the criteria specified by the International League Against Epilepsy (ILAE) [12]. The EEGs of patients with rolandic spikes, with or without clinical seizures, were reviewed by pediatric neurologists (M.-L.T. and F.-C.L.) for this research, and the diagnosis of children with rolandic spikes without clinical seizures was based on EEG findings, with variable indications for EEG studies, including headache, dizziness, Tourette syndrome, attention-deficit hyperactivity disorder (ADHD), learning disability, enuresis, or developmental delay. Data retrieved from EEG registration were reviewed and identified. All patients were enrolled from the outpatient clinic and had undergone regular follow-up at our institutions. Patients with abnormal MRIs, brain tumors, vascular malformations, neurocutaneous diseases, malformation of cortical development, and patients with atypical BECTS, continuous spike-waves during slow-wave sleep (CSWS), and Landau–Kleffner syndrome were excluded from this study. 

The patients were divided into 2 groups and compared with age-matched healthy subjects: BECTS with seizures (*n* = 31), BECTS (rolandic spikes) without clinical seizures (*n* = 21), and age-matched controls (*n* = 20). The age-matched control group comprised children who visited our outpatient clinics for various conditions, such as anxiety, or for routine health examinations that were unrelated to neurologic or psychiatric illness. The inclusion criteria for the age-matched controls included children with no neurological disorders, no previous loss of consciousness, and no family history of a first-degree relative with epilepsy or febrile seizures. Medical records were reviewed and clinical parameters, including age, gender, age at onset of seizures, seizure frequency (total episodes), duration of seizures, and the presence of secondary generalized seizures, were recorded. Furthermore, comorbidities that are associated with developmental delays, such as autism, ADHD, cognitive delays, learning disability, or other neurologic symptoms were analyzed. 

### 2.2. EEG Examinations and Epoch Selection 

All of the EEGs were drug-naïve and within 1 year after seizures or symptoms onset. The EEGs were recorded by a 32-channel digital EEG system (Nihon Kohden Neurofax EEG-1200J, Japan or Natus Nicolet EEG wireless 64 Amplifier, Middleton, WI, USA) with a sample rate of at least 200 Hz. Impedances were less than 10 kΩ. The international 10–20 system was used for electrode placement, and a total of 19 scalp electrodes were placed. Two electrodes were used to record horizontal and vertical ocular artifacts (electrooculogram (EOG)). EEGs were recorded for at least 50 min in the awake and natural sleep states without sedation in all patients. Data were recorded from common reference electrodes placed on the earlobes, and the EOG was recorded.

Raw EEGs were transformed into European Data Format (EDF) files with bandpass filters of between 0.5 and 70 Hz and a 60 Hz notch filter. Data were referenced to the average values obtained from all channels. The selected epoch was a 60 s artifact-free section characterized by the absence of epileptiform discharges or patterns of drowsiness in the resting-awake state. Awake EEG with overabundant epileptiform discharges and unable to select a stable spike-free epoch for over 6 s were excluded because the post-spike period may interfere with EEG background activity [13]. Only the presence of continuous physiologic alpha rhythms with maximum over posterior regions without mechanical or physiological artifacts were selected.

### 2.3. EEG Data Processing 

We first imported continuous scalp EEG data into the MATLAB-based open toolbox EEGLAB (version 2019.0) [14]. Artifacts resulting from eye movements, blinking, and other mechanical effects were removed via independent component analysis (ICA) decomposition [15] before partitioning the EEG data into 29 epochs, each of which was 4 s in duration with overlaps of 2 s. The epochs were carefully examined to ensure that they were free from signals that corresponded to muscular contractions, such as head movement. We then sought to identify functional networks across broad regions of the brain by estimating the functional connection strength based on the phase-locking values (PLVs) obtained from all electrodes (Fp1, Fp2, F3, F4, C3, C4, P3, P4, O1, O2, F7, F8, T3, T4, T5, T6, Fz, Cz, and Pz). The frequency bands in this analysis were as follows: alpha (8–13 Hz), beta (13–30 Hz), delta (0.5–4 Hz), and theta (4–8 Hz).

### 2.4. Functional Connectivity 

The PLVs were employed in the construction of functional networks between all 19 electrode pairs for each frequency band and epoch pair. To reduce the influence of volume conduction, efforts were made to preserve phase synchronization [16]. PLVs can be used to quantify the degree of asymmetry in the distribution of instantaneous phase differences [16]. For example, symmetric distributions centered around zero are indicative of spurious connectivity, whereas a flat distribution indicates a complete lack of connectivity, and deviations from a symmetric distribution are indicative of intrasource dependency. PLVs range between 0 and 1, where 0 indicates an absence of coupling, and 1 indicates perfect phase locking. We produced weighted adjacency matrices for each measure of connectivity by averaging all 29 matrices of functional connectivity for each patient with each corresponding frequency band.

### 2.5. Analysis Based on Graph Theory 

We constructed weighted connectivity matrices by applying thresholds to 19 × 19 weighted adjacency matrices for each subject and the corresponding frequency band. Setting the thresholds using the 90th, 85th, 80th, …, and 10th percentiles of the weights resulted in 17 matrices that ranged in density from 10% to 90%, in increments of 5%. Indices based on graph theory were then used to analyze the weighted connectivity matrices [17]. In graph theory, brain models comprise nodes (representing EEG channels) with undirected edges (representing functional connections obtained using PLV). In this study, graphing parameters were obtained using the Brain Connectivity Toolbox (https://sites.google.com/site/bctnet/ accessed on 28 August 2021). We obtained the following four indices for graph reconstruction [18]: (1) basic nodal strength values, (2) measures of integration as an indicator of global efficiency, (3) measures of segregation based on a clustering coefficient, and (4) measures of betweenness centrality.

### 2.6. Statistical Analysis 

Categorical variables are displayed as numbers and percentages and were analyzed with the Chi-square test or Fisher’s exact test (when the expected cell number was less than 5); the Mann–Whitney U test and the *t*-tests for continuous variables were used for intergroup comparisons as appropriate. The two-sided Wilcoxon rank-sum test was used to compare every 2 groups in graphs of the theoretic properties, and the results were later assessed after correcting for multiple comparisons using the false discovery rate (FDR) [19]. For correlation between age of onset of seizures, seizure frequency, and each parameter of EEG functional connectivity, the rank partial correlation coefficient was plotted on graph indices that were statistically significant. All statistical analysis was performed using the MATLAB R2021a environment (MathWorks Inc., Natick, MA, USA). A *p* < 0.05 indicated statistically significant differences, unless otherwise indicated.

## 3. Results

### 3.1. Clinical Characteristics and EEG Findings

A total of 52 BECTS patients, diagnosed according to EEG findings, were included in this study, after the exclusion of patients with abnormal MRIs, those who were lost to follow-up, and those with poor EEG recording quality or with too many epileptiform discharges during the awake state. Children with BECTS diagnosed based on EEGs were categorized into two groups: (1) BECTS with seizures (*n* = 31, 19 boys and 12 girls; age, mean ± SD = 8.45 ± 2.59, range 4.0–12.1 years); and (2) BECTS without clinical seizures (*n* = 21, 11 boys and 10 girls; age, mean ± SD = 8.59 ± 2.46, range 5.1–14 years). Twenty children were enrolled in the age-matched control group (12 boys and 8 girls; age, mean ± SD = 8.65 ± 2.82, range 5.2–12.1 years). The unpaired t-test and the Chi-square test did not show statistically significant intergroup differences in age or gender among the three subgroups (Table 1). 

In children who had BECTS with clinical seizures, the total seizure frequency before medication or without medication was less than or equal to 2 times in 15 patients, 3–5 times in 13 patients, and more than or equal to 6 times in 3 children; the average seizure frequency before EEGs was 2.90 ± 1.68. Five patients had focal to bilateral tonic–clonic seizures, according to their medical records. Seven patients had a history of febrile seizures (FS), including three with simple recurrent FS, three with simple single FS, and one with complex FS. Three patients had a family history of epilepsy, and two had a family history of FS. With regard to comorbidities in the BECTS with seizures group, 16 (48.4%) patients had ADHD or ADD, 4 were previously diagnosed with mild or borderline developmental delay, 3 (9.7%) had headache, 1 (3.2%) had enuresis, 1 (3.2%) had ASD trait, 1 (3%) had learning disability, and 1 (3.2%) had a previously diagnosed tic disorder. In children who had rolandic spikes without clinical seizures, 9 (42.9%) had ADHD or ADD; 8 (38.1%) had headache, migraine, or dizziness; 7 (33.3%) had tic disorder or Tourette syndrome, 4 (19%) had previously diagnosed borderline or mild developmental delay, 1 (4.8%) had ASD trait, and 2 (9.5%) had learning disability (Table 1). 

All studied EEG records were analyzed with no medication and within one year of seizure onset. Among the 52 patients with rolandic spikes, all the sleep EEGs had a horizontal dipole of negative epileptiform discharges over the centrotemporal area and a positive dipole over the frontal or midline areas. In the BECTS with seizures group, 13 (41.9%) patients had an EEG focus on the right side, 8 (25.8%) on the left side, and 10 (32.2%) had bilateral independent foci, compared to the BECTS without clinical seizures group, where 4 (19%) patients had the EEG focus on the right side, 5 (23.8%) on the left side, and 12 (57.1%) had bilateral independent foci (Table 1).

### 3.2. Functional Connection Strength Based on the PLVs

Figure 1 demonstrates the edge connection strength among the three groups in four frequency bands. Compared to controls, the BECTS with seizures group had significantly higher connectivity strength, which was maximal in the midline central, right frontal, right central, and temporal regions in the theta frequency band, and was maximal in the inferior frontal and anterior temporal region in the alpha frequency band (Figure 1A). Furthermore, the connectivity strength was significantly higher in the BECTS patients without clinical seizures compared to controls across almost all electrodes in the theta band, and was also evident in electrodes in the alpha and beta frequency bands (Figure 1B). Only very few instances of decreased connection strength in the delta frequency band between the BECTS patients with or without seizures were evident compared to age-matched controls. There was no significant difference in connectivity strength between BECTS patients with seizures and BECTS patients without seizures in four frequency bands (Figure 1C). These results were all corrected for multiple comparisons with the FDR.

### 3.3. EEG Functional Connectivity Findings Using Graph Theory Analysis

The network properties of EEGs were measured with the following indices: (1) global efficiency, (2) mean clustering coefficient, (3) mean betweenness centrality, and (4) mean nodal strength in four frequency bands (delta, theta, alpha, and beta). The proportional thresholds were set to between 0.1 and 0.9 of the weights in the matrix.

We found that BECTS patients with seizures have significantly higher global efficiency (≥40% proportional threshold), mean clustering coefficient (≥55% proportional threshold), and mean nodal strength (≥35% proportional threshold) in theta frequency band compared to age-matched controls (Figure 2, adjusted *p* < 0.05, based on the two-sided Wilcoxon rank-sum test after correction by FDR). In addition, BECTS patients with seizures have significantly lower mean betweenness centrality in theta, alpha, and beta frequency bands mostly at high thresholds (>60% proportional threshold) compared to age-matched controls (adjusted *p* < 0.05, Figure 2). BECTS patients without seizures also have significantly higher global efficiency (≥20% proportional threshold), mean clustering coefficient (≥30% proportional threshold), and mean nodal strength (≥10% proportional threshold) in the theta frequency band at almost all thresholds, and higher global efficiency in the beta band at almost all thresholds (≥20% proportional threshold) compared to age-matched controls (*p* < 0.05, in the two-sided Wilcoxon rank-sum test after correction by FDR; Figure 3); however, there was no statistically significant difference in mean betweenness centrality compared to age-matched controls. There was no statistically significant difference in all four types of network properties (global efficiency, mean clustering coefficient, mean betweenness centrality, and mean nodal strength) in all four frequency bands between BECTS patients with and without clinical seizures (Appendix A).

### 3.4. Correlations between Age of Onset, Seizure Frequency, and EEG Functional Connectivity

Seizure frequency, age of onset of seizures, or age of onset of symptoms for EEG examination were correlated with four EEG network properties, including global efficiency, mean clustering coefficient, mean betweenness centrality, and mean nodal strength in all four frequency bands in each proportional threshold (from 10 to 90% in increments of 5%). A significantly positive correlation was found between the age of onset of seizures and global efficiency, mean clustering coefficient, mean nodal strength in the alpha frequency band (adjusted *p* < 0.01 at ≥35% proportional threshold), mean clustering coefficient, mean nodal strength in the theta frequency band, and mean nodal strength in the delta frequency band in BECTS patients with seizures (Figure 4). Moreover, a positive significant correlation was found between symptom onset and global efficiency, mean clustering coefficient, and mean nodal strength in the beta frequency band at almost all proportional thresholds (adjusted *p* < 0.05 at 0.1, 0.15, and ≥0.25 proportional thresholds) in BECTS patients without clinical seizures (Figure 4). However, there were no statistically significant differences between seizure frequency and all measured functional connectivity in the BECTS with seizures group. No statistically significant correlation was found between age of seizure onset and seizure frequency in the BECTS with seizures group (r = −0.15, *p* = 0.43). 

## 4. Discussion

Our findings indicate that children with rolandic spikes with or without clinical seizures presented with disrupted resting-state EEG connectivity in comparison with age-matched controls, and had significantly higher global efficiency, mean clustering coefficient, and mean nodal strength in the theta frequency band compared to age-matched controls. However, the mean betweenness centrality was significantly lower in theta, alpha, and beta frequency bands in BECTS children with seizures, but not in those without seizures compared to age-matched controls. The connectivity strength between channels was significantly higher in children with BECTS with or without seizures, and had increased connectivity strength between spatially distant channels within or across hemispheres in the theta frequency band (Figure 1). An increased global efficiency likely reflects increased long-range connectivity and oversynchronization at the whole-brain level, and higher mean clustering coefficients might indicate increased short-range connectivity in the theta band. The decreased function of mean betweenness centrality was present in BECTS patients with seizures, but not in those without clinical seizures. This finding may suggest that weaker local connectivity may induce a lower seizure threshold. 

One study using surface EEG showed that ripples (80–250 Hz) on rolandic spikes were present mostly in atypical BECTS and BECTS with seizures, compared to patients without seizures [20]. We did not include patients with atypical BECTS and CSWS because we considered these diseases were different from BECTS in terms of the pathogenesis, disease courses, and prognosis. Thus, ours is the first known study to investigate EEG functional connectivity in children with rolandic spikes without clinical seizures, which are common in pediatric neurology practice. Therefore, this electrophysiological finding may be used as a tool for clinical assessment.

### 4.1. Neuronal Synchrony, Global Efficiency, Clustering Coefficient, and Nodal Strength Were Higher in BECTS Children with or without Clinical Seizures

Global efficiency and nodal strength indicate the integration of functional networks, whereas clustering coefficients reflect the segregation function. Previous fMRI research showed that the epileptic brain has higher global efficiency due to the increased number of functional connections between networks globally, and the decreased local efficiency in between-network connections in the cinguloopercular task-control network [21]. Increased theta synchronization is often observed in epilepsy or epilepsy presenting with brain tumors [22,23]. Furthermore, EEG functional connectivity studies in patients with temporal lobe epilepsy demonstrated significantly increased synchronization in the theta band, but not in the alpha band [24], with which our results match. High coherence across multiple frequencies in all regions beyond the centrotemporal areas was inferred in other research, indicating the strong network connectivity between brain regions in BECTS patients [25]. Excessive brain connectivity may be as harmful as underconnectivity because neuronal synchrony is finely tuned in the brain and serves in the adaptation of behavior and cognition [26].

The results of EEG connectivity in BECTS have been quite variable. In BECTS patients, Adebimpe et al. found increased synchronization in certain brain regions close to the epileptogenic zones as well as decreased synchronization in other regions characterized by the absence of epileptic spikes [27]. Another report showed increased EEG functional connectivity in the beta band globally [13]. In contrast, some studies found decreased functional connectivity in BECTS patients [28,29]. The clustering coefficient represents clusters of edges that connect neighboring nodes and indicates a measure of network segregation. In a systematic review and meta-analysis, patients with focal epilepsy displayed a higher interictal clustering coefficient in the whole brain [30]. The nodal strength is the sum of the weight and links of a node. The nodal strengths were related to global efficiency, clustering coefficient, edge, and threshold. Our results showed higher connectivity in global efficiency, mean clustering coefficient, and mean nodal strength in BECTS patients with or without clinical seizures compared to controls at the whole-brain level.

### 4.2. Betweenness Centrality Was Lower in BECTS Children with Seizures, but Not in Those without Seizures

Betweenness centrality is a measure of centrality in a graph based on the shortest paths in graph theory. Research has shown that nodes with high betweenness centrality are potential hubs for surgical resection [31], and may be used as an indicator to monitor seizure reduction after ketogenic diet therapy in childhood epilepsy [32]. Betweenness centrality might indicate localized connectivity and potentially plays a protective role in seizure spread or recurrence [32]. Our results showed decreased function of betweenness centrality in the theta, alpha, and beta frequency bands in BECTS patients with seizures, but not in those without clinical seizures compared to controls, and might indicate weaker local connectivity in BECTS patients with clinical seizures.

### 4.3. Higher Connection Strength (Edge) in BECTS with or without Seizures Compared to Controls

The connectivity strengths between nodes were based on graph measures and allowed for assessment of the organization of the brain network function. The connectivity strength between channels in the majority was significantly higher in BECTS patients with or without seizures compared with controls, specifically in the theta frequency bands after FDR correction (Figure 1). The connectivity strengths were higher between most of the spatially distant channels within and between hemispheres in the theta band in the BECTS without seizures group; however, no statistically significant difference was found between the seizure and nonseizure groups. Some stronger connections between adjacent channels might be attributable to the volume conduction of electric signals. We found maximum strong connectivity more frequently over the right fronto-centro-temporal areas in BECTS patients with seizures, and this finding might be due to the presence of a higher number of epileptogenic foci on the right side (42%). Widespread increased connectivity strength in the theta frequency band in BECTS patients without seizures might be due to the mostly bilateral epileptogenic foci (57%). Research showed increased neuronal coupling between frontal and temporal regions in bilateral beta band connectivity [13]; moreover, increased EEG functional connectivity within epileptogenic zones and other hemispheres has been interpreted as compensatory effort in epileptic patients [33].

### 4.4. Age of Onset Is Positively Associated with EEG Functional Connectivity

With regard to the relationship between clinical parameters, the age of onset of seizures was positively associated with the strength of EEG functional connectivity, including global efficiency, mean clustering coefficient, and mean nodal strength in the delta, theta, and alpha frequency bands. In patients who were younger at the age of onset of seizures or other symptoms, a significant loss of functional connectivity was found at the time of EEG recording. Early seizure onset impacts the focal connectivity in the hippocampus in TLE, even in the absence of MTS [34]. It has been reported that age was positively associated with functional connectivity, including global efficiency, clustering coefficient, and path length, and was negatively correlated with age with a logarithmical pattern in epileptic children [35,36]. However, we did not find a correlation between age of seizure onset and frequency of seizures. 

### 4.5. Limitations

Although enrolled BECTS children were all premedication and drug-naïve, the evolution of the EEGs in BECTS needs to be carefully evaluated. We controlled for this confounder using age-matched analysis, used age as a confounder in correlation analysis of clinical parameters, and FDR-corrected all statistical results. No chloral hydrate or sedative was given during the EEG study. Furthermore, we used standard EEG electrodes instead of highly dense EEG recording because of the retrospective study design.

## 5. Conclusions

In conclusion, similar to BECTS patients with seizures, children with rolandic spikes without clinical seizures presented with excessive neuronal synchrony, global efficiency, clustering coefficient, nodal strength, and connectivity strengths, specifically in the theta frequency band. Weaker local connectivity presented with significantly lower betweenness centrality only in patients with clinical seizures, and may further lower the seizure threshold. The difference in local connectivity may be used as a tool for EEG functional connectivity for clinical assessment. As rolandic spikes existed in the learning stage in children, future large-scale prospective studies on the correlation of neuropsychological performance and functional connectivity as well as clinical trials of effective treatment options are needed.

## Figures and Tables

**Figure 1 biomedicines-10-01553-f001:**
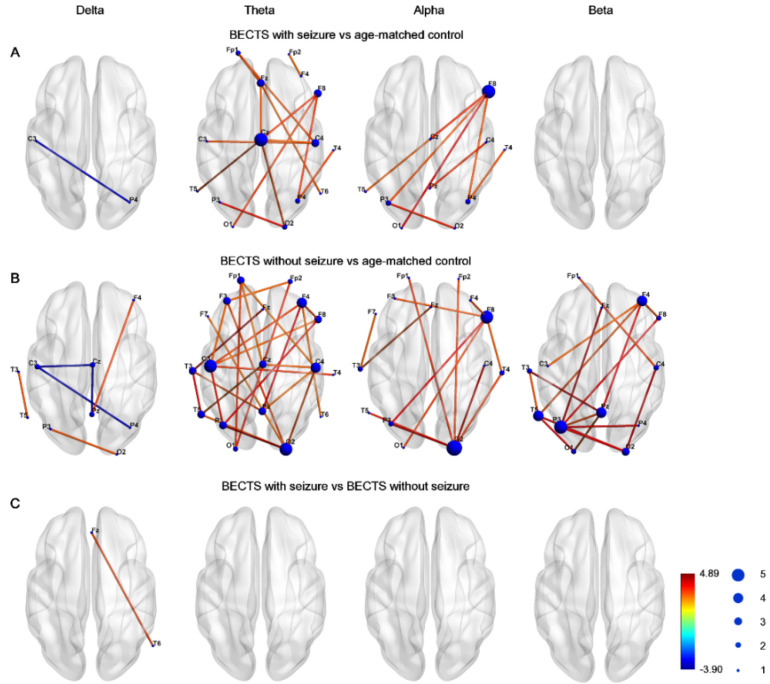
Graphs demonstrated network–based statistics of edges in different frequency bands in (**A**) BECTS with seizures, (**B**) BECTS without seizures, and (**C**) age–matched control groups. Line color represented *t* value between each pair of channels across all subjects in each group. The size of node represented frequency of significance in each node. Note there was higher strength between BECTS with or without seizures compared to age-matched controls in the theta band.

**Figure 2 biomedicines-10-01553-f002:**
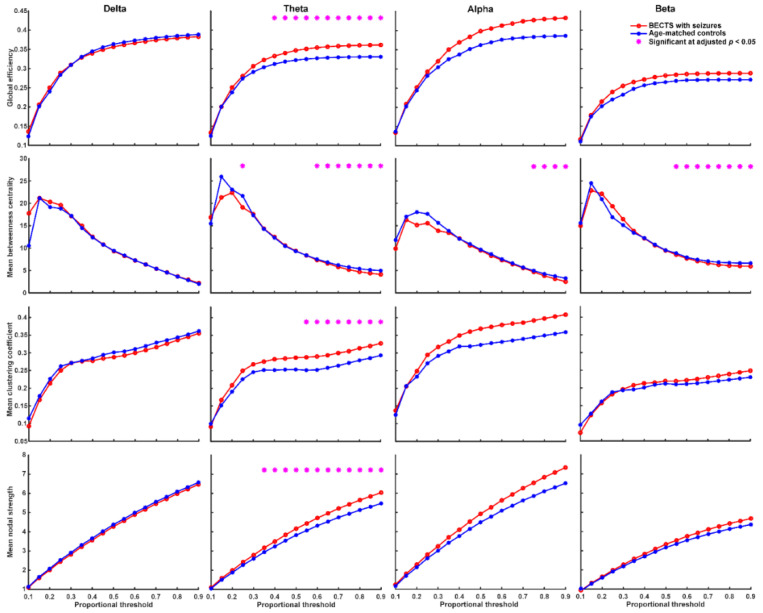
Graph-based functional connectivity indices between BECTS with seizures and age-matched controls as each proportional threshold at each frequency band. BECTS children with seizures have significantly higher global efficiency (≥40% proportional threshold), mean clustering coefficient (≥55% proportional threshold), and mean nodal strength (≥35% proportional threshold) in the theta band compared to age-matched controls (* adjusted *p* < 0.05). BECTS children with seizures have lower mean betweenness centrality in theta, alpha, and beta bands mostly at high thresholds compared to age-matched controls.

**Figure 3 biomedicines-10-01553-f003:**
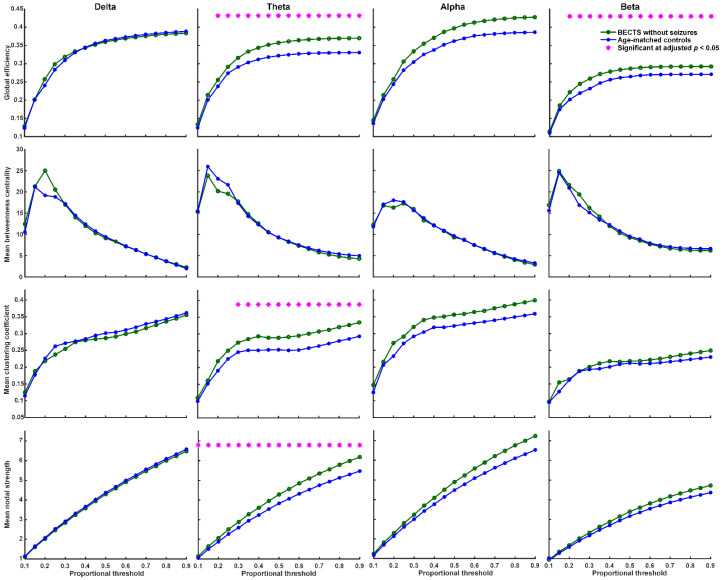
Graph-based functional connectivity indices between BECTS without clinical seizures and age-matched controls as each proportional threshold at each frequency band. Children with BECTS without clinical seizures have significantly higher global efficiency (≥20% proportional threshold), mean clustering coefficient (≥30% proportional threshold), and mean nodal strength (≥10% proportional threshold) in the theta band and higher global efficiency in the beta band at almost all proportional thresholds (≥20%) compared to age-matched controls (* adjusted *p* < 0.05).

**Figure 4 biomedicines-10-01553-f004:**
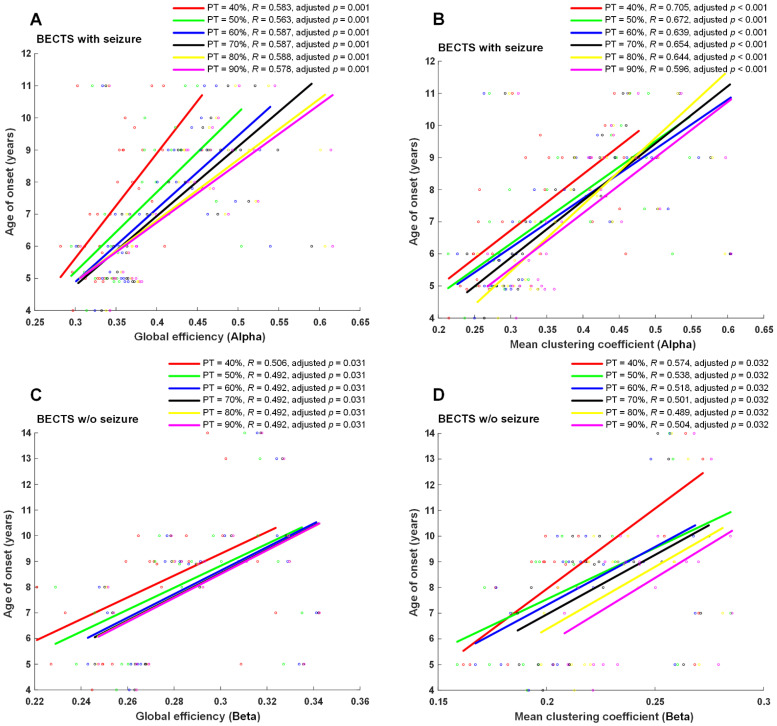
Scatter plots and correlation of graph-theory-based analysis vs. age of onset. Age of onset of seizures was significantly positively correlated with global efficiency and mean clustering coefficients in the alpha band (adjusted *p* ≤ 0.01 at ≥40% proportional threshold), shown in (**A**,**B**) in BECTS children with seizures. Additionally, a positive significant correlation was found between onset of symptoms and global efficiency and mean clustering coefficient in the beta band at almost all proportional thresholds (adjusted *p* < 0.05 at ≥40% proportional threshold), shown in (**C**,**D**) in BECTS children without seizures. w/o: without.

**Table 1 biomedicines-10-01553-t001:** Clinical demographics in children with benign childhood epilepsy with centrotemporal spikes (BECT) with or without behavior seizures before treatment.

	Controls	BECT with Seizures (no. of Patients)	BECT without Seizures (no. of Patients)	*p* Value
Patient number	20	31	21	
Gender (male%)	12 (60.0%)	19 (61.3%)	11 (52.4%)	0.802 (Chi-square)
Age at EEG (years old, mean ± SD)	8.65 ± 2.82	8.45 ± 2.59 years (range 4.0–12.1)	8.59 ± 2.46 years (range 5.1–14)	0.962 (ANOVA, f = 0.04)
Age of onset (years old, mean ± SD)	–	7.33 ± 2.10 years (range 3.8–11)	8.08 ± 2.65 years (range 4.0–14.0) (other symptoms)	0.262 (*t*-test, *t* = −1.134)
Family history of epilepsy or febrile seizures	–	5	0	
Febrile seizures history	0	7 (22.5%)	0	
Seizure frequency before EEG	–	2.90 ± 1.68	-	
Focal to FBCT	–	5 (16.1%)	0	
**Comorbidities**				
ADHD/ADD		16 (48.4%)	9 (42.9%)	
History of developmental delay		4 (12.9%)	4 (19.0%)	
Enuresis		1 (3.2%)	1 (4.8%)	
Tic disorder/Tourette disease		1 (3.2%)	7 (33.3%)	
Learning disability		1 (3.2%)	2 (9.5%)	
ASD trait		1 (3.2%)	1 (4.8%)	
Headache/dizziness		3 (9.7%)	8 (38.1%)	
**EEG foci**				
Right		13 (41.9%)	4 (19.0%)	N.S.
Left		8 (25.8%)	5 (23.8%)	N.S.
Bilateral		10 (32.2%)	12 (57.1%)	N.S.

ADHD/ADD: attention-deficit hyperactivity disorder/attention-deficit disorder; ASD: autistic spectrum disorder; FBTCS: focal to bilateral tonic–clonic seizures; SD: standard deviation; Unpaired *t*-test, ANOVA, or nonparametric Mann–Whitney (M-W) two-sample test/Chi-square test/Fisher exact test where appropriate. N.S.: not significant.

## Data Availability

Not applicable.

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
