# Peer review of "Resting-State EEG Functional Connectivity in Children with Rolandic Spikes with or without Clinical Seizures"

_biomedicines, 2022, doi:10.3390/biomedicines10071553_

Round 1

Reviewer 1 Report

The authors presented a retrospective study where they analyzed resting state EEG in terms of functional connectivity in BECTS patients with and without clinical seizures. The results were compared to those of age and sex matched healthy controls.

The main result of the study was that patients with BECTS, irrespective of the presence of clinical seizures, differ from the control group in terms of various components of functional connectivity. The only difference between the two patient groups was that the decreased mean betweenness centrality was only present in BECTS patients with seizures as compared to those without clinical seizures.

The paper is well written. My main concerns are related to some conclusions which are not supported by the results of the study (see below).

Minor points:

Methods.

Line 100: “The patients were divided in 3 groups”… one of them is a control group where essentially healthy subjects were included. I would suggest writing that the patients were divided into 2 groups and compared with age and sex matched healthy subjects.

Neuropsychology was performed only in 16 patients with seizures – I would completely get rid of the section related to neuropsychology from methods, results and discussion as it does not contribute to paper.

Line 178: Chi-square – should be written with a capital “C”

Line 135: Exclusion of patients with “too many” spikes during wakefulness is an unclear statement. Please, be more precise.

Discussion:

Line 348: The suggestion of utilizing resting state EEG functional connectivity as a clinical biomarker based only on a single retrospective study with a relatively small cohort is exaggerated. Please, rephrase this sentence also in conclusions.

Conclusion:

 “Disrupted functional connectivity may adversely affect cognition, attention, and learning in children with rolandic spikes even if they do not present with clinical seizures. As rolandic spikes existed in the learning stage in children, these findings may affect the standard treatment policy in BECTS patients with or without clinical seizures”. The results of this study by no means could lead to such conclusions. Please, remove this section.

Reviewer 2 Report

The Authors of this paper investigated extremely important issues concerning EEG functional connectivity in children with rolandic spikes.

The research and results are described in great detail, but the introduction lacks a description of the individual EEG parameters that were studied (lines 254-256). Which entails their increase or decrease. In particular, that changes in these parameters are referred to in the discussion (subsection 4.1.).

Section 4.4 - the description lacks information as to what are the consequences of the observations described regarding the influence of age on seizure frequency and also, for example, on the prognosis as to their disappearance. Will it be possible to determine on the basis of an EEG result that seizures may occur despite previous symptoms? In this paper, too little emphasis is placed on the practical application of the results obtained. 
